# Infrared Thermography as a Potential Non-Invasive Tool for Estrus Detection in Cattle and Buffaloes

**DOI:** 10.3390/ani13081425

**Published:** 2023-04-21

**Authors:** Umair Riaz, Musadiq Idris, Mehboob Ahmed, Farah Ali, Liguo Yang

**Affiliations:** 1International Joint Research Centre for Animal Genetics, Breeding and Reproduction (IJRCAGBR), Huazhong Agricultural University, Wuhan 430070, China; umair.riaz@iub.edu.pk (U.R.); mehboob.alyani@gmail.com (M.A.); 2Key Laboratory of Animal Genetics, Breeding and Reproduction, Ministry of Education, College of Animal Science and Technology, Huazhong Agricultural University, Wuhan 430070, China; 3Frontiers Science Center for Animal Breeding and Sustainable Production, Huazhong Agricultural University, Wuhan 430070, China; 4Department of Theriogenology, Faculty of Veterinary and Animal Sciences, The Islamia University of Bahawalpur, Bahawalpur 63100, Punjab, Pakistan; drfarahiub@gmail.com; 5Department of Physiology, Faculty of Veterinary and Animal Sciences, The Islamia University of Bahawalpur, Bahawalpur 63100, Punjab, Pakistan; musadiq.idris@iub.edu.pk; 6Livestock and Dairy Development Department, Lahore 54000, Punjab, Pakistan

**Keywords:** infrared thermography, estrus detection, cattle, buffalo

## Abstract

**Simple Summary:**

Estrus detection is the most important factor for dairy animals (cattle and buffaloes) to get them pregnant on time for efficient production. Over a period of time, increased production potential caused increased stress and weakened estrus expression in cattle. While buffaloes have known problems with silent heat, the existing conventional methods of estrus detection are somewhat labor intensive and less efficient. Similarly, the modern automated methods that rely on detecting physical activity are expensive, and their efficiency is affected by factors such as type of housing (tie stall), flooring, and environment. Infrared thermography has recently emerged as a technique that does not depend on monitoring physical activity. Furthermore, it is a non-invasive, user-friendly technique that requires relatively less physical handling of animals. Hence, infrared thermography possesses great potential to be used for generating estrus alerts in dairy animals for efficient breeding and consequent production.

**Abstract:**

The productivity of dairy animals has significantly increased over the past few decades due to intense genetic selection. However, the enhanced yield performance of milk animals caused a proportional increase in stress and compromised reproductive efficiency. Optimal reproductive performance is mandatory for the sustainable production of dairy animals. Reproductive efficiency is marked by proper estrus detection and precise breeding to achieve maximum pregnancies. The existing conventional methods of estrus detection are somewhat labor intensive and less efficient. Similarly, the modern automated methods that rely on detecting physical activity are expensive, and their efficiency is affected by factors such as type of housing (tie stall), flooring, and environment. Infrared thermography has recently emerged as a technique that does not depend on monitoring physical activity. Furthermore, infrared thermography is a non-invasive, user-friendly, and stress-free option that aids in the detection of estrus in dairy animals. Infrared thermography has the potential to be considered a useful non-invasive tool for detecting temperature fluctuations to generate estrus alerts without physical contact in cattle and buffaloes. This manuscript highlights the potential use of infrared thermography to understand reproductive physiology and practical implementation of this technique through discussing its advantages, limitations, and possible precautions.

## 1. Introduction

Substantial efforts have been made over the past few decades to improve the productivity of milk [1] and meat [2] of farm animal species. This emphasis is mandatory to uplift the socio-economic status of livestock farmers and to address the convincing global threat to food security [3] and stunting growth for the ever-escalating world population [4,5]. The strategy of genetic selection for production traits was fruitful to increase milk yield. However, enhanced production capabilities resulted in a proportional increase in stress on the dairy animals, compromising their reproductive efficiency. Furthermore, due to the negative genetic correlation with reproductive traits, the long-term selection for productive traits has resulted in decreased reproductive performance in dairy animals [6,7,8].Optimal reproductive performance is mandatory for sustainable production. Efficient estrus detection and breeding at the optimal time to achieve higher pregnancies are practical goals of reproductive management [9]. Therefore, various invasive, laborious, and expensive interventions were brought into practice to address the reproductive issues [10].

Continuous scientific efforts are in place around the world to develop efficient, low-cost, user-friendly techniques for estrus detection. Recently, infrared thermography has emerged as a technique to aid in estrus detection with efficiency without compromising animal welfare [11]. Therefore, the objective of this review is to explain the use of IRT to understand reproductive physiology, its advantages, disadvantages, precautions, and its practical implementation.

## 2. Issues of Estrus Expression in Cattle and Buffalo

Cattle and buffalo are the major contributors to the world’s milk supply (Table 1) as both species produce 96% of the total milk produced [12]. The duration of estrus and its intensity has been compromised corresponding to an increase in production potential, whereas the buffalo is known for its less pronounced sexual behavior and poor expression of estrus. Therefore, an efficient estrus detection strategy is mandatory to maintain the production cycle of dairy animals.

Genetic selection has been effectuated to increase the per-head production of dairy animals [6] in order to mitigate management inputs and the burden on agricultural land. Data summarized by some studies indicates a substantial increase in milk corresponding to a decrease in the number of animals [13]. On the contrary, increased production resulted in enhanced stress on the food-producing animals, jeopardizing their reproductive efficiency [14,15]. Rearte et al. [16] published data on cattle in Argentina (from 2000 to 2012) highlighting the negative effect of milk production on reproductive efficiency. Negligible inheritance of reproductive traits remained a limiting factor to resolve this issue with genetic selection [17]. It is a well-established fact that for efficient livestock farming, the reproductive performance of animals must be stellar, as it is paramount in achieving maximum production in terms of meat and milk [18,19]. Therefore, provided with limited chances for genetic selection, management interventions become very significant to improve reproductive efficiency and to harvest maximum potential from the animal of superior production genetics.

Estrus detection and insemination/breeding of dairy animals at the right time are pivotal in achieving reproductive goals [20,21]. To defy the challenge of diminishing reproductive performance, corresponding to production stress, various strategies (estrus synchronization, fixed time artificial insemination, embryo transfer, etc.) have been developed for farm animals to control estrous physiology and precise breeding [22,23].However, they are labor-intensive, invasive, expensive, and stressful for animals. These may result in snagging farm operations, increasing input cost, and raising animal welfare issues. Therefore, efforts are being made to develop cost-effective, handy, and unstrained techniques for animals. Infrared thermography (IRT) is a noninvasive technique that is being used in veterinary practice to diagnose various ailments in animals [24]. It detects changes in surface temperature from a safe distance and aids in detecting estrus-related thermal changes with more efficiency than conventional methods. Additionally, IRT is cost-effective and easy to implement in farm conditions [25], and has been studied to validate its use for estrus detection in dairy animals.

## 3. Importance of Estrus Expression

Estrus detection is by far the most important event at the dairy farm. It is the only time during the 21-day long estrous cycle when a female animal can breed to become pregnant and eventually start production. Failure to spot estrus or its faulty detection leads to no breeding or insemination at a less optimal time [26]. This results in low fertility, long calving intervals, and ultimately less calving. Therefore, farm productivity and economics are negatively affected [27]. Hence, proper estrus detection and insemination at a precise time is mandatory for successful dairy livestock farming. Faulty detection of estrus of dairy animals could result in serious economic losses (Figure 1) due to reduced reproductive efficiency, low production, and increased managemental cost [9].

## 4. Existing Methods of Estrus Detection

Estrus is characterized by several behavioral signs under the influence of estrogen. The primary sign of estrus is ‘stands to be mounted’ by other animals in the herd. Other secondary signs of estrus include restlessness, bellowing, frequent urination, swelling, and redness of the vulva along with a rise in body temperature. These changes in the behavior of animals are recorded to breed the animal at the optimal time to maximize the chances of pregnancy. The best time to inseminate the cow is 12 h after the onset of standing heat, well known as the “a.m./p.m.-p.m./a.m.” rule [28]. Similarly, the best insemination time in buffaloes is reported to be 24 h after the onset of standing heat because of delayed ovulation [29] compared to cows [30]. Similar relations, in terms of time interval with respect to the occurrence of secondary signs of estrus and optimal time of insemination, are established in farm animals [31,32].

Keeping in view this importance, various means are employed at dairy farms for estrus detection, ranging from visual observations to the installation of automated equipment at the farm.

A brief account of these methods is given below.

### 4.1. Visual Observation

Identification of animals exhibiting estrus via a visual observation by trained personnel is the most widely practiced method [33]. It requires the observing person to have enough knowledge of estrus signs and behaviors. This method requires multiple observations of the herd in a day to identify animals in heat.

To identify animals in standing heat, mounting behavior is observed. Cows usually show homosexual behavior as females mount on each other during or just around the estrus along with other secondary signs. Contrarily, buffaloes lack such homosexual behavior and other estrus signs are also vague compared to cows [34]. Therefore, in such condition, introduction of a penile deviated or vasectomized male is almost mandatory in buffalo herds for accurate estrus detection. Similarly, introducing a male in cow herds could be beneficial as it enhances the chances of animals in estrus and its intensity [35]. Secondary signs are also helpful in identifying animals in heat [36]. Animals exhibiting estrus behavior are recorded and should be confirmed by examining the status of the reproductive system through rectal palpation before subjecting them to artificial insemination.

Only up to 50% of animals are accurately detected in estrus by visual observation [37]. A decrease in estrus duration and intensity over a period has been reported and it has further affected the efficacy of visual observation of estrus [26,36]. Some aids in visual detection, such as chin ball markers and pressure-sensitive mount detection devices (Kamar Heatmount Detectors; Kamar Products Inc., Zionsville, IN, USA), could be helpful to an extent. The output of this method could be increased by enhancing the number of observations per day, but it requires to incur more labor and cost [38].

### 4.2. Camera-Assisted Estrus Detection

To overcome the labor load and increase the rate of estrus detection, video camera-assisted surveillance of the dairy herd is performed. This can be accomplished by day-night monitoring of the herd and sorting out the hyperactive animals exhibiting estrus signs. In another method, specialized pressure-sensitive devices are passed on to the sacrococcygeal region of the cow. Under the pressure exerted by the bull while mounting, the color in the device is changed. This extent of color variation is detected by a suitably placed camera installed with particular software [39,40]. The rate of estrus detection using this technique increased and it is advantageous because of the enhanced sensitivity, specificity, and predictive value of the technique [40]. Limiting factors of this method include improper identification of animals and displacement of the device itself, or rubbing with other objects in the vicinity. This option is available to farmers at a high installation cost; this factor limits its use by most of them [38]. There is no doubt in the efficiency of the method, however, the performance of camera-assisted estrus detection can be enhanced by replacing this camera with a thermal camera with an added advantage of its potential capability to assess estrus detection through thermal variation.

### 4.3. Activity Monitors

A marked increase in activity is observed in animals during estrus [41]. Cashing in on this behavioral exploitation, activity monitors can be engaged for efficient estrus detection without extending labor inputs. The sensors are usually applied to the neck or legs of the animal. These sensors quantify the resting, standing, and walking duration, as well as neck movements [42]. Activity monitors have an automated signaling system and generate alerts if variation in activity is observed over a specified period of time. Various studies have analyzed the association between the time of enhanced activity during estrus, its respective time of ovulation, and fertility [43,44]. The success of an activity monitor depends upon its make and software. The threshold of increased intensity installed to detect estrus is a major factor in determining the efficiency of these instruments. There is reluctance among farmers to adopt these systems due to the initial high cost of installation [38].

### 4.4. Measurement of Milk Progesterone

Progesterone from the corpus luteum of the ovary is a very reliable endocrine indicator of cycling animals. In non-pregnant animals, its concentration is found higher during the luteal phase. Luteolysis and the consequent drop in progesterone concentration are mandatory before the next estrus [45]. This signature variation in the progesterone profile can be used to predict estrus or ovulation. Progesterone concentrations can be measured in the blood plasma, but for large animals it is not a suitable option as blood collection is highly stressful. The alternative non-invasive option is measuring the progesterone in milk [46], but it is very laborious to collect, label, and process milk samples of an individual animal in a large herd. Automatic progesterone analyzers integrated with milking machines are also available that draw a curve of progesterone in milk and an alert is given when it starts decreasing below the threshold level [47]. However, due to the high installation cost of such progesterone detecting systems, dairy farmers are reluctant to adopt it instead of conventional methods.

### 4.5. Detection of Changes in Body Temperature

Reproductive cyclicity in dairy animals is characterized by interdependent sequential physiological events denoted by changes in ovarian dynamics, endocrine profile, behavioral response, and body temperature [48]. It has been observed that body temperature drops two days before estrus, while a sudden rise is observed just before the LH peak [49]. This temperature rise may be attributed to increased activity or endocrine profile around this particular period of the estrous cycle [50]. This fluctuation could be used to estimate the time of ovulation to fruitful inseminations. Several studies utilize changes in body temperature to predict estrus in dairy animals using different techniques [32,51,52,53]. There are chances of false positive cases using this method that can be minimized using a good manual or automated professional software-based breeding record of the herd.

## 5. Thermal Changes during the Estrous Cycle

Rhythmic changes in the body temperature are an important and convenient indicator to assess health status and energy metabolism during different physiological events [48]. The estrous cycle is characterized by rhythmic alterations in the endocrine profile, behavior, and physical activity. These recurrent variations are the outcome of follicular as well as luteal dynamics and related hormonal variations. Correspondingly, the animal’s body temperature exhibits similar changes during various physiological events of the estrous cycle [48]. These thermal variations congruent to estrus physiology have been assessed through the rectum, vagina, and body surfaces in farm animals [51,54,55].

Temperature variations of the animal’s body during the estrous cycle are reported to be attributed to the increase in physical activity by some scientists [56]. A higher concentration of estrogen hormone is manifested behaviorally with hyperactivity. Estrogen increases peripheral blood supply and enhances heat dissipation from the surface, which can be detected. It dilates blood vessels by acting on vascular smooth muscle and triggering the release of nitric oxide, which is an endothelial vasodilator [57]. This relationship of physical activity with temperature rise during estrus was negated when thermal fluctuation during estrus was studied in tied animals. The rise in body temperature was observed even though their activity was limited [52,58]. It highlights the possibility that hyperactivity is not the single reason for increased body temperature and connotes the presence of some thermogenic mechanism independent of physical activity during estrus. On the other hand, some reports suggest that such detectable thermal fluctuations in the animal’s body are linked to progesterone concentrations [51,59]. The role of progesterone as a thermogenic agent has been well-defined in whales [60] and humans [61]. Higher body temperature was observed when progesterone was given exogenously to ovariectomized human [62] and rat [63] females. The exact mechanism by which progesterone brings on the thermogenic effect is not well understood. A rise in body temperature was observed in a higher progesterone environment during the luteal phase in cows. However, Lammoglia et al. [58] narrated that no direct relation exists between progesterone concentration and body temperature in cattle. Elevation in body temperature appears to be associated with periovulation junctures. Temperature spikes are considered more related to a preovulatory LH surge [55]. The interval from temperature rise to LH varies among different studies, however, the variability of this duration is persistent in indicating a strong interrelationship [64]. The increase in body temperature associated with estrus ranges from 0.5 °C to 1.3 °C [48,58]. The difference in variation may be influenced by managemental practices, housing conditions, environment, and the methods adopted to detect body temperature. Ovulation takes place after a specific interval corresponding to the LH surge; similarly, there is a correlation between temperature rise and LH surge. Therefore, the detection of fluctuation in body temperature has the potential to generate alerts preceding the LH surge and ovulation [55,64].

Suthar et al. [52] summarized the pattern of temperature variations during estrus, such as body temperature rise during estrus, drops around ovulation followed by a rise during diestrus, and another fall two to three days before the onset of estrus [58].

## 6. Infrared Thermography as a Tool for Estrus Detection

Changes in surface temperature during various physiological stages relevant to reproduction in farm animals have been studied intensively in the recent past using IRT [51,59,65]. IRT cameras are capable of detecting minute changes in temperatures [66]. This makes IRT a very useful tool for detecting these changes in surface temperature associated with physiological events related to reproduction in animals.

### 6.1. Working Principle and Application

Radiant heat is generated within the infrared portion of the electromagnetic spectrum by every object present on earth. Anything with a temperature above zero kelvin (absolute temperature = −273.15 °C) radiates electromagnetic waves [67]. Living organisms, especially homeotherms, emit heat through the body’s surface to maintain homeostasis. Fluctuations in surface temperature could be attributed to different physiological statuses and metabolic activities [68]. During thermoregulation, the skin acts as an organ of heat exchange through evaporation. Similarly, changes in vascularization and blood flow during different physiological conditions result in changes in skin temperature [69]. These phenomena make the skin surface an ideal organ to assess thermal alterations. Temperature measurements through body surfaces are highly correlated with rectal temperature and the former could be the alternative to the latter [59]. Thermographic cameras are helpful in detecting heat radiation and even the occurrence of minute thermal fluctuations [70]. Thermographic sensors in the cameras detect the heat radiation and generate a picture of varying pixels as illustrated in Figure 2. Thermographs present different colors based on temperature variation from the radiation being emitted from any surface and that can be processed through computer-based software. These varying colors of thermographs are interpreted and digitized for exact measurements of temperature variations [25]. Infrared thermography, as a tool to measure body temperature, is reported to be better and more advantageous compared to other methods. Furthermore, manual determination of body temperature is time-consuming, practically difficult for repeated or continuous observations, and involves the risk of disease transmission [25].

### 6.2. Potential of Infrared Thermography for Estrus Detection

Infrared thermography offers the advantage of being operated from a specified distance without physical interaction with animals. Therefore, it causes minimum stress to animals and also ensures their welfare during the assessment of various essential parameters [11]. Additionally, this technique is user-friendly, easy to handle, and has less operating cost [25]. Infrared thermography is used in various industrial operations [71] and as a diagnostic tool in medical [72] and veterinary practices [73]. As far as dairy animal farming is concerned, this technology is proven to be effective in diagnosing heat stress [74], mastitis [75], postpartum diseases [76], and lameness [77]. It has also been reported to assess feed intake and reproductive status in livestock [78]. The adoption of infrared thermography in livestock farming is considered to be an advanced, sensitive, cost-effective, rapid, and non-invasive technique that does not require stressful physical interaction with animals.

The advancement in infrared thermography made it possible to assess body changes in body temperature during the estrous cycle with minute physical strain on animals from various anatomical surfaces (Vulva, eye, muzzle, etc.) [51,58,59]. Emissivity is the ability of a surface to emit heat and it is scored on a scale from 0 to 1. The emissivity of an object is highest (1) or lowest (0) depending upon the amount of heat being omitted. The cattle body surface possesses very good emissivity that varies from 0.93 to 0.98 depending on the density and color of hairs and skin [79]. Therefore, the skin surface of animals serves as an organ of great value to detect temperature variation through infrared thermography. Furthermore, infrared thermography has great potential for automation compared to conventional methods of temperature measurement [80]. Previously, different techniques were exercised to correlate fluctuations in body temperature during estrus with behavioral, endocrinal, and structural changes in the reproductive tract during estrus in animals. Most of these studies involved the measurement of body temperature through the rectum and vagina via invasive techniques [81,82]. Such physical handling of animals could cause stress and uneasiness, altering the body temperature and risking the authenticity of results. All the above-mentioned practical conveniences and technical advantages highlight the great potential of IRT to be implemented in livestock operations, including reproductive management or estrus detection. Infrared thermography is proven to have the promising potential to predict estrus in order to minimize calving interval and increase the number of pregnant animals per artificial insemination [51]. Various studies conducted in dairy cows, beef cattle, and buffaloes to correlate events related to estrous physiology with variations in body temperature using thermography have been summarized in Table 2. These studies have highlighted that infrared thermography could be an important tool, alone or combined with other techniques, to detect estrus in dairy animals.

Animals exhibit rhythmic variations in body temperature [74]. These thermal changes are dependent on physical activity or progesterone concentrations during the estrous cycle. Infrared thermography has the potential to detect thermal fluctuations related to different physiological states during the estrous cycle.

### 6.3. Important Consideration for the Use of Infrared Thermography

Implementation of infrared thermography in reproductive management requires the consideration of some important factors that can affect its efficiency. These factors are related to the animal under observation, environment, and operator [73].

The factors related to animals include the coloration of skin surface and hair, the density of hairs, and unwanted motions at the time of observations. Hairs provide insulation to the skin surface affecting its emissivity [87]. Therefore, usually hairless areas of the body such as the muzzle, eye, and vulva are preferred for thermography measurements. As far as skin coloration is concerned, buffaloes and black-colored cattle are believed to have a greater emissivity because of the ability of black color to absorb and emit more infrared radiation compared to lighter colors [88]. Similarly, hair coloration has a corresponding effect and dark-colored hairs have greater IRT value compared to those of lighter colors [89]. The presence of dirt or moisture on the surface under observation can affect IRT readings [70].

From the environmental point of view, ambient temperature and humidity at the time of observation are important factors that can influence IRT measurements [90]. Similarly, the IRT readings can vary when taken at different times (morning, noon, temperature) on the same day [91]. This could be due to variations in meteorological indicators (sunlight, temperature, and humidity) or due to changes in physical activity.

In the morning, before feeding and milking, animals have minimum physical activity compared to other times in the day when extra activity-dependent heat is generated and emitted through the skin surface [92]. In the relevant scientific Literature, recommendations are found to take IRT readings in the morning in the absence of sunlight and activity. Furthermore, IRT readings taken below 30 °C or 20 °C have been reported to reduce environmental impact [93,94].The altered physiological response of the animals at noon, especially during hot days [95], may be the possible reason for preferring morning time IRT recording.

Operator-dependent factors of IRT include distance and angle between the thermographic camera and emissive surface. If the animal is made to walk to bring it to the observation place, it is better to provide an acclimatization time of about 10 min before taking the IRT reading. It is essential because peripheral blood circulation increases during activity, which enhances heat dissipation through the skin [75].

## 7. Conclusions

Different techniques of estrus detection are in place in a tool kit for estrus detection, with well-reported advantages and limitations of each technique. Infrared thermography, especially the computer assisted automation of IRT systems, could be helpful in generating estrus alerts and has a potential to be added as one of the useful techniques in the tool kit of estrus detection techniques.

## Figures and Tables

**Figure 1 animals-13-01425-f001:**
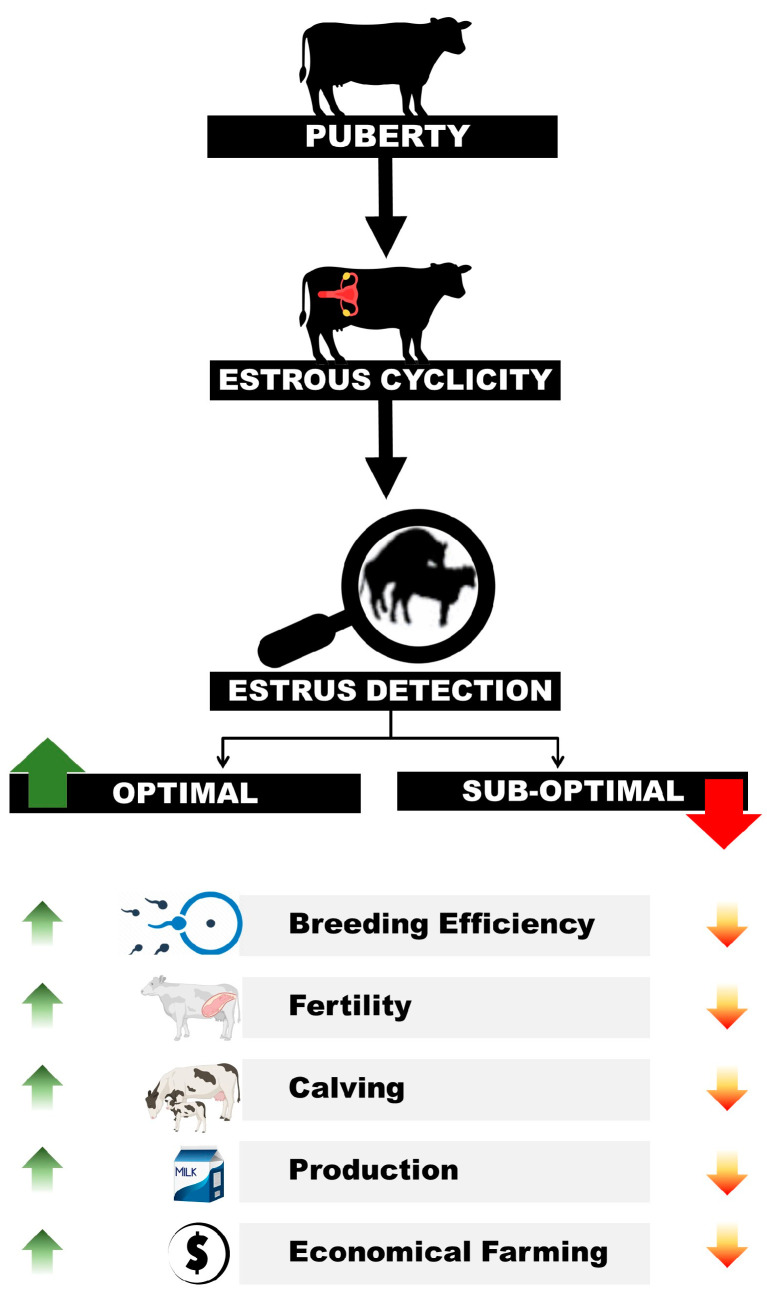
Illustrative diagram of the economic impact of optimal and suboptimal estrus detection in dairy animals.

**Figure 2 animals-13-01425-f002:**
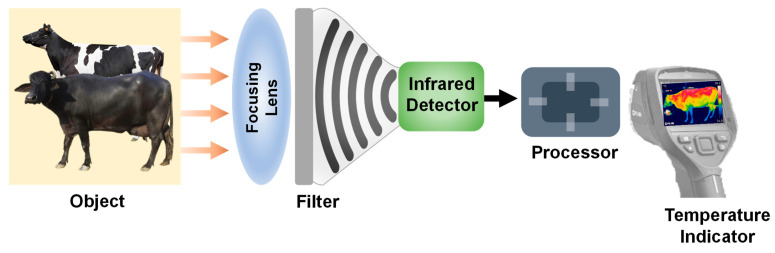
Illustrative diagram of the working principle of an infrared thermographic camera.

**Table 1 animals-13-01425-t001:** World’s milk production share of dairy animals (Source: https://www.fao.org/dairy-production-products/en/, accessed on 5 March 2023).

Sr. No.	Dairy Animal	World’s Milk Production Share
1	Cattle	81%
2	Buffalo	15%
3	Goat	2%
4	Sheep	1%
5	Camel	0.5%
6	Other species	0.5%

**Table 2 animals-13-01425-t002:** Studies on application of Infrared Thermography (IRT) to detect estrus in cattle and buffaloes.

Sr. No.	References	Study Animal	Site of IRT Observation	Conclusion
1	Marquez et. al., 2022 [83]	Dairy cows	Vulva	Automated IRT platform has the potential to become an alternative to visual estrus detection.
2	Tiwari et. al.,2022 [84]	Sahiwal cows	Muzzle and vulva	Infrared thermos radiography may be used as an efficient tool for the detection of estrus and its different stages in Sahiwal cows.
3	Rajput et. al., 2022 [85]	Sahiwal cows	Vulval, eyeball, ear and muzzle	IRT is an upcoming non-invasive technology that can be used to monitor increases in temperature of Sahiwal cows during estrus.
4	Marquez et. al., 2021 [11]	Dairy cows	Vulva	The combination of thermal and behavioral parameters increased the accuracy of estrus detection.
5	Vicentini et. al.,2020 [53]	Dairy Heifers	Eye, vulva, and muzzle	In conclusion, IRT is an effective method to detect temperature variation during the proestrus and estrus phases in Gyr heifers.
6	Marquez et. al., 2019 [86]	Dairy cows	Eye, muzzle, cheek, neck, front right foot, front left foot, rump, flank, vulva area, tail head, and withers	Fluctuations in radiated temperature measured at specific anatomical locations and the frequency of tail movements and treading behaviors can be used as a noninvasive estrus alert in multiparous cows housed in a tie stall system.
7	Deak et. al., 2019 [24]	Dairy cows	ocular globe, muzzle, pelvis, abdomen, thorax, perineum, mammary gland	Season and reproductive phases influence the surface temperature of body areas.
8	Ruediger et. al., 2018 [59]	Buffaloes	vulvar, orbital area and muzzle	The vulvar superficial temperature is effective in ascertaining the physiological changes inherent to the progesterone concentrationvariation during the reproductive cycle.
9	Radigonda et. al., 2017 [65]	Braford cows	vulvar	IRT as an indirectly diagnostic tool to detect ovarian activity seems promising and further studies are required to validate their potential in beef cattle production.
10	Talukder et. al., 2014 [51]	Dairy cows	Vulva and muzzle	IRT possesses potential to aid in estrus detection.

## Data Availability

Not applicable.

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
