# Peer review of "Infrared Thermography as a Potential Non-Invasive Tool for Estrus Detection in Cattle and Buffaloes"

_animals, 2023, doi:10.3390/ani13081425_

Round 1
Reviewer 1 Report
General comments:
The goal of this manuscript is to review the current methods of estrus detection and to compare the use of infrared thermography as an alternative technology for detecting estrus. While this is a valuable technology, some of the arguments made in the manuscript are not necessarily correct.
Most of your arguments in favor of using infrared thermography instead of other methods are based on this technology not requiring handling of animals and the animal welfare concerns involved with it. However, the most common methods of estrus detection (i.e. visual detection, estrus detection patches) do not involve handling of animals as well. Furthermore, I do not agree that estrous synchronization protocols, the use of pedometers, or sensors will pose an animal welfare issue. Handling animals is part of livestock production, specially in the dairy industry, and does not necessarily pose an animal welfare issue (i.e. The amount of stress the cow will experience when handled to have e pedometer installed is likely the same amount of stress that she will have when handled for milking).
In addition, you argue that proper detection of estrus is crucial for the reproductive management and to optimize fertility of dairy cows. Yet, you fail to compare the efficiency in estrus detection and pregnancy rates of using thermography in comparison to other traditional methods. Does thermography provide a more accurate detection of estrus? If so, does it result in improved fertility when compared to other methods?
Furthermore, how close does the technician have to be from the animal in order to get an accurate read of the temperature? Would the animals be in the pasture 30 meters away from the technician, or does that require closing the animals in a smaller pen in order to get an accurate read?
Specific comments:
Line 26: Milk instead of milch
Line 32-33: You claim the use of artificial insemination and embryo transfer to involve stressful physical handling of animals. While thermography allows detection of estrus without stressful handling of animals, you would still need to perform artificial insemination or embryo transfer in these animals, so your technology does not decrease any stressful physical handling of animals.
Line 59-60: You say "duration of estrus and its intensity has been compromised" what caused this?
Line 61: "is known for" rather than "is known its"
Line 61: What do you mean by buffaloes having shy behavior?
Line 66: This will sound better if you remove the "since" from the beginning of the sentence.
Line 70: You say "increased production resulted in enhanced stress" why? how?
Line 77: I suggest limited instead of meager
Lines 85-86: why are they risky? Millions of animals are bred this way every year without complications.
I think that you would drive a better argument if you were to focus on the time, labor, and monetary investment that those technologies require instead of the stress or the risk for the operator and the animal. Livestock production will always require handling of animals
Line 91: I am not sure of what you mean with "being used in veterinary practice to diagnose various aliments in animals"
Line 92: As you detect the animal in estrus, you will still have to handle the animal to perform artificial insemination. Furthermore, estrus detection can be done visually without handling animals, or with the aid of estrus detection patches. You argue that this technology will allow for less handling of animals, which is not necessarily true.
Line 93: I suggest implement instead of handle
Line 101: I suggest "will be negatively affected" instead of "got shattered"
Line 126: I suggest "vocalization and frequent urination" instead of "bellowing, frequent micturition" (It's just easier for the common reader)
Line 167: The color in the device rather than the color is the device
Line 249: Not sure what you mean with "the detection of thrombogenicity"
Line 280: I suggest time consuming rather than time taking
Lines 290-292: These are not required for estrus detection
Lines 298-299: So does visual observation and use of estrus detection patches
Line 327: "Infrared thermography is proven to have promising potential" instead of "Infrared thermography is proven to be having the promising potential"
Line 332: Table 2, not table 1
Line 355: meteorological instead of metrological
Lines 359-361: What if the estrus related changes in body temperature happen at noon? How would the temperature and activity of animals affect the capacity to detect estrus using this technique?
Author Response
Authors are again grateful to the reviewer for his contribution to spare time to review this very important manuscript. The valuable suggestion and comments of the reviewer have helped us to improve the manuscript further before getting it published.
The detailed Response is attached as a separate file.

Reviewer 2 Report
The manuscript is in general well written, but needs extensive English review. Specially when it comes to standardized English. One can notice different patterns of English - British, American, Indian and so on.
Lines 45 - 46 - It is important to mention that productive and reproductive traits usually have negative genetic correlation (not only caused by stress). It means some genes that are responsible for the expression of those trais have pleiotropic effects. The authors should include this information.
Table 1. I believe you should cite the source of this information again in the Table title.
As a review paper, altough it seems to be very important, I believe it is very basic. The author should provide more practil examples in order to make the manuscript publishable
Author Response
Authors are again grateful to the reviewer for his contribution to spare time to review this very important manuscript. The valuable suggestion and comments of the reviewer have helped us to improve the manuscript further before getting it published.
The detailed response to the reviewer 2 is attached as a separate file.

Round 2
Reviewer 2 Report
I believe the manusript is ready to be published in Animals